# The Blinded Bandit:
# Learning with Adaptive Feedback

**Ofer Dekel**
Microsoft Research
oferd@microsoft.com

**Elad Hazan**
Technion
ehazan@ie.technion.ac.il

**Tomer Koren**
Technion
tomerk@technion.ac.il

## Abstract

We study an online learning setting where the player is temporarily deprived of feedback each time it switches to a different action. Such model of *adaptive feedback* naturally occurs in scenarios where the environment reacts to the player's actions and requires some time to recover and stabilize after the algorithm switches actions. This motivates a variant of the multi-armed bandit problem, which we call the *blinded multi-armed bandit*, in which no feedback is given to the algorithm whenever it switches arms. We develop efficient online learning algorithms for this problem and prove that they guarantee the same asymptotic regret as the optimal algorithms for the standard multi-armed bandit problem. This result stands in stark contrast to another recent result, which states that adding a switching cost to the standard multi-armed bandit makes it substantially harder to learn, and provides a direct comparison of how feedback and loss contribute to the difficulty of an online learning problem. We also extend our results to the general prediction framework of bandit linear optimization, again attaining near-optimal regret bounds.

## 1   Introduction

The *adversarial multi-armed bandit problem* [4] is a $T$-round prediction game played by a randomized player in an adversarial environment. On each round of the game, the player chooses an *arm* (also called an *action*) from some finite set, and incurs the loss associated with that arm. The player can choose the arm randomly, by choosing a distribution over the arms and then drawing an arm from that distribution. He observes the loss associated with the chosen arm, but he does not observe the loss associated with any of the other arms. The player's *cumulative loss* is the sum of all the loss values that he incurs during the game. To minimize his cumulative loss, the player must trade-off *exploration* (trying different arms to observe their loss values) and *exploitation* (choosing a good arm based on historical observations).

The loss values are assigned by the adversarial environment before the game begins. Each of the loss values is constrained to be in $[0, 1]$ but otherwise they can be arbitrary. Since the loss values are set beforehand, we say that the adversarial environment is *oblivious* to the player's actions.

The performance of a player strategy is measured in the standard way, using the game-theoretic notion of *regret* (formally defined below). Auer et al. [4] present a player strategy called EXP3, prove that it guarantees a worst-case regret of $O(\sqrt{T})$ on any oblivious assignment of loss values, and prove that this guarantee is the best possible. A sublinear upper bound on regret implies that the player's strategy improves over time and is therefore a *learning* strategy, but if this upper bound has a rate of $O(\sqrt{T})$ then the problem is called an *easy*[1] online learning problem.

In this paper, we study a variant of the standard multi-armed bandit problem where the player is temporarily *blinded* each time he switches arms. In other words, if the player's current choice is different than his choice on the previous round then we say that he has *switched* arms, he incurs the loss as before, but he does not observe this loss, or any other feedback. On the other hand, if the player chooses the same arm that he chose on the previous round, he incurs and observes his loss as usual[2]. We call this setting the *blinded multi-armed bandit*.

For example, say that the player's task is to choose an advertising campaign (out of $k$ candidates) to reduce the frequency of car accidents. Even if a new advertising campaign has an immediate effect, the new accident rate can only be measured over time (since we must wait for a few accidents to occur) and the environment's reaction to the change cannot be observed immediately.

The blinded bandit setting can also be used to model problems where a switch introduces a temporary bias into the feedback, which makes this feedback useless. A good example is the well-known *primacy and novelty effect* [14, 15] that occurs in human-computer interaction. Say that we operate an online restaurant directory and the task is to choose the best user interface (UI) for our site (from a set of $k$ candidates). The quality of a UI is measured by the the time it takes the user to complete a successful interaction with our system. Whenever we switch to a new UI, we encounter a primacy effect: users are initially confused by the unfamiliar interface and interaction times artificially increase. In some situations, we may encounter the opposite, a novelty effect: a fresh new UI could intrigue users, increase their desire to engage with the system, and temporarily decrease interaction times. In both cases, feedback is immediately available, but each switch makes the feedback temporarily unreliable.

There are also cases where switching introduces a variance in the feedback, rather than a bias. Almost any setting where the feedback is measured by a physical sensor, such as a photometer or a digital thermometer, fits in this category. Most physical sensors apply a low-pass filter to the signal they measure and a low-pass filter in the frequency domain is equivalent to integrating the signal over a sliding window in the time domain. While the sensor may output an immediate reading, it needs time to stabilize and return to an adequate precision.

The blinded bandit setting bears a close similarity to another setting called the adversarial multi-armed bandit with *switching costs*. In that setting, the player incurs an additional loss each time he switches arms. This penalty discourages the player from switching frequently. At first glance, it would seem that the practical problems described above could be formulated and solved as multi-armed bandit problems with switching costs and one might question the need for our new blinded bandit setting. However, Dekel et al. [12] recently proved that the adversarial multi-armed bandit with switching costs is a *hard* online learning problem, which is a problem where the best possible regret guarantee is $\widetilde{\Theta}(T^{2/3})$. In other words, for any learning algorithm, there exists an oblivious setting of the loss values that forces a regret of $\widetilde{\Omega}(T^{2/3})$.

In this paper, we present a new algorithm for the blinded bandit setting and prove that it guarantees a regret of $O(\sqrt{T})$ on any oblivious sequence of loss values. In other words, we prove that the blinded bandit is surprisingly as easy as the standard multi-armed bandit setting, despite its close similarity to the hard multi-armed bandit with switching costs problem. Our result has a theoretical significance and a practical significance. Theoretically, it provides a direct comparison of how feedback and loss contribute to the difficulty of an online learning problem. Practically, it identifies a rich and important class of online learning problems that would seem to be a natural fit for the multi-armed bandit setting with switching costs, but are in fact much easier to learn. Moreover, to the best of our knowledge, our work is the first to consider online learning in an setting where the loss values are oblivious to the player's past actions but the feedback is adaptive.

We also extend our results and study a blinded version of the more general bandit linear optimization setting. The bandit linear optimization framework is useful for efficiently modeling problems of learning under uncertainty with extremely large, yet structured decision sets. For example, consider the problem of online routing in networks [5], where our task is to route a stream of packets between two nodes in a computer network. While there may be exponentially many paths between the two nodes, the total time it takes to send a packet is simply the sum of the delays on each edge in the path. If the route is switched in the middle of a long streaming transmission, the network protocol

needs a while to find the new optimal transmission rate, and the delay of the first few packets after the switch can be arbitrary. This view on the packet routing problem demonstrates the need for a blinded version of bandit linear optimization.

The paper is organized as follows. In Section 2 we formalize the setting and lay out the necessary definitions. Section 3 is dedicated to presenting our main result, which is an optimal algorithm for the blinded bandit problem. In Section 4 we extend this result to the more general setting of bandit linear optimization. We conclude in Section 5.

## 2   Problem Setting

To describe our contribution to this problem and its significance compared to previous work, we first define our problem setting more formally and give some background on the problem.

As mentioned above, the player plays a $T$-round prediction game against an adversarial environment. Before the game begins, the environment picks a sequence of loss functions $\ell_1, \ldots, \ell_T : \mathcal{K} \mapsto [0, 1]$ that assigns loss values to arms from the set $\mathcal{K} = \{1, \ldots, k\}$. On each round $t$, the player chooses an arm $x_t \in \mathcal{K}$, possibly at random, which results in a loss $\ell_t(x_t)$. In the standard multi-armed bandit setting, the feedback provided to the player at the end of round $t$ is the number $\ell_t(x_t)$, whereas the other values of the function $\ell_t$ are never observed.

The player's expected cumulative loss at the end of the game equals $\mathbb{E}[\sum_{t=1}^{T} \ell_t(x_t)]$. Since the loss values are assigned adversarially, the player's cumulative loss is only meaningful when compared to an adequate baseline; we compare the player's cumulative loss to the cumulative loss of a fixed policy, which chooses the same arm on every round. Define the player's *regret* as

$$R(T) \;=\; \mathbb{E}\left[\sum_{t=1}^{T} \ell_t(x_t)\right] \;-\; \min_{x \in \mathcal{K}} \sum_{t=1}^{T} \ell_t(x) \;. \tag{1}$$

Regret can be positive or negative. If $R(T) = o(T)$ (namely, the regret is either negative or grows at most sublinearly with $T$), we say that the player is *learning*. Otherwise, if $R(T) = \Theta(T)$ (namely, the regret grows linearly with $T$), it indicates that the player's per-round loss does not decrease with time and therefore we say that the player is not learning.

In the *blinded* version of the problem, the feedback on round $t$, i.e. the number $\ell_t(x_t)$, is revealed to the player only if he chooses $x_t$ to be the same as $x_{t-1}$. On the other hand, if $x_t \neq x_{t-1}$, then the player does not observe any feedback. The blinded bandit game is summarized in Fig. 1.

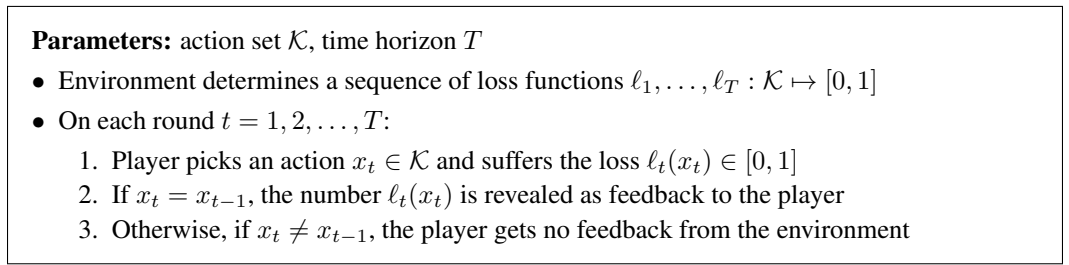

**Parameters:** action set $\mathcal{K}$, time horizon $T$
- Environment determines a sequence of loss functions $\ell_1, \ldots, \ell_T : \mathcal{K} \mapsto [0, 1]$
- On each round $t = 1, 2, \ldots, T$:
    1. Player picks an action $x_t \in \mathcal{K}$ and suffers the loss $\ell_t(x_t) \in [0, 1]$
    2. If $x_t = x_{t-1}$, the number $\ell_t(x_t)$ is revealed as feedback to the player
    3. Otherwise, if $x_t \neq x_{t-1}$, the player gets no feedback from the environment

Figure 1: The blinded bandit game.

**Bandit Linear Optimization.**   In Section 4, we consider the more general setting of online linear optimization with bandit feedback [10, 11, 1]. In this problem, on round $t$ of the game, the player chooses an action, possibly at random, which is a point $x_t$ in a fixed action set $\mathcal{K} \subset \mathbb{R}^n$. The loss he suffers on that round is then computed by a linear function $\ell_t(x_t) = \ell_t \cdot x_t$, where $\ell_t \in \mathbb{R}^n$ is a loss vector chosen by the oblivious adversarial environment before the game begins. To ensure that the incurred losses are bounded, we assume that the loss vectors $\ell_1, \ldots, \ell_T$ are *admissible*, that is, they satisfy $|\ell_t \cdot x| \leq 1$ for all $t$ and $x \in \mathcal{K}$ (in other words, the loss vectors reside in the *polar set* of $\mathcal{K}$). As in the multi-armed bandit problem, the player only observes the loss he incurred, and the full loss vector $\ell_t$ is never revealed to him. The player's performance is measured by his regret, as defined above in Eq. (1).

## 3 Algorithm

We recall the classic EXP3 algorithm for the standard multi-armed bandit problem, and specifically focus on the version presented in Bubeck and Cesa-Bianchi [6]. The player maintains a probability distribution over the arms, which we denote by $p_t \in \Delta(\mathcal{K})$ (where $\Delta(\mathcal{K})$ denotes the set of probability measures over $\mathcal{K}$, which is simply the $k$-dimensional simplex when $\mathcal{K} = \{1, 2, \ldots, k\}$). Initially, $p_1$ is set to the uniform distribution $(\frac{1}{k}, \ldots, \frac{1}{k})$. On round $t$, the player draws $x_t$ according to $p_t$, incurs and observes the loss $\ell_t(x_t)$, and applies the update rule

$$\forall\, x \in \mathcal{K}, \quad p_{t+1}(x) \; \propto \; p_t(x) \cdot \exp\left(-\eta\, \frac{\ell_t(x_t)}{p_t(x_t)} \cdot \mathbb{1}_{x=x_t}\right) .$$

EXP3 provides the following regret guarantee, which depends on the user-defined learning rate parameter $\eta$:

**Theorem 1** (due to Auer et al. [4], taken from Bubeck and Cesa-Bianchi [6])**.** *Let $\ell_1, \ldots, \ell_T$ be an arbitrary loss sequence, where each $\ell_t : \mathcal{K} \mapsto [0, 1]$. Let $x_1, \ldots, x_T$ be the random sequence of arms chosen by* EXP3 *(with learning rate $\eta > 0$) as it observes this sequence. Then,*

$$R(T) \; \leq \; \frac{\eta k T}{2} + \frac{\log k}{\eta} \;\; .$$

EXP3 cannot be used in the blinded bandit setting because the EXP3 update rule cannot be called on rounds where a switch occurs. Also, since switching actions $\Omega(T)$ times is, in general, required for obtaining the optimal $O(\sqrt{T})$ regret (see [12]), the player must avoid switching actions too frequently and often stick with the action that was chosen on the previous round. Due to the adversarial nature of the problem, randomization must be used in controlling the scheme of action switches.

We propose a variation on EXP3, which is presented in Algorithm 1. Our algorithm begins by drawing a sequence of independent Bernoulli random variables $b_0, b_1, \ldots, b_{T+1}$ (i.e., such that $\mathbb{P}(b_t = 0) = \mathbb{P}(b_t = 1) = \frac{1}{2}$). This sequence determines the schedule of switches and updates for the entire game. The algorithm draws a new arm (and possibly switches) only on rounds where $b_{t-1} = 0$ and $b_t = 1$ and invokes the EXP3 update rule only on rounds where $b_t = 0$ and $b_{t+1} = 1$. Note that these two events can never co-occur. Specifically, the algorithm always invokes the update rule one round before the potential switch occurs. This confirms that the algorithm relies on the value of $\ell_t(x_t)$ only on non-switching rounds.

---
**Algorithm 1:** BLINDED EXP3
---

set $p_1 \leftarrow (\frac{1}{k}, \ldots, \frac{1}{k})$, draw $x_0 \sim p_1$
draw $b_0, \ldots, b_{T+1}$ i.i.d. unbiased Bernoullis

**for** $t = 1, 2, \ldots, T$
    **if** $b_{t-1} = 0$ *and* $b_t = 1$
        draw $x_t \sim p_t$                              `// possible switch`
    **else**
        set $x_t \leftarrow x_{t-1}$                                  `// no switch`

    play arm $x_t$ and incur loss $\ell_t(x_t)$

    **if** $b_t = 0$ *and* $b_{t+1} = 1$
        observe $\ell_t(x_t)$ and for all $x \in \mathcal{K}$, update

$$w_{t+1}(x) \; \leftarrow \; p_t(x) \cdot \exp\left(-\eta\, \frac{\ell_t(x_t)}{p_t(x_t)} \cdot \mathbb{1}_{x=x_t}\right)$$

        set $p_{t+1} \leftarrow w_{t+1}/\|w_{t+1}\|_1$
    **else**
        set $p_{t+1} \leftarrow p_t$

---

We set out to prove the following regret bound.

**Theorem 2.** *Let $\ell_1, \ldots, \ell_T$ be an arbitrary loss sequence, where each $\ell_t : \mathcal{K} \mapsto [0,1]$. Let $x_1, \ldots, x_T$ be the random sequence of arms chosen by Algorithm 1 as it plays the blinded bandit game on this sequence (with learning rate fixed to $\eta = \sqrt{\frac{2 \log k}{kT}}$). Then,*

$$R(T) \leq 6\sqrt{Tk \log k} \ .$$

We prove Theorem 2 with the below sequence of lemmas. In the following, we let $\ell_1, \ldots, \ell_T$ be an arbitrary loss sequence and let $x_1, \ldots, x_T$ be the sequence of arms chosen by Algorithm 1 (with parameter $\eta > 0$). First, we define the set

$$S = \left\{ t \in [T] \ : \ b_t = 0 \text{ and } b_{t+1} = 1 \right\} \ .$$

In words, $S$ is a random subset of $[T]$ that indicates the rounds on which Algorithm 1 uses its feedback and applies the EXP3 update.

**Lemma 1.** *For any $x \in \mathcal{K}$, it holds that*

$$\mathbb{E}\left[ \sum_{t \in S} \ell_t(x_t) - \sum_{t \in S} \ell_t(x) \right] \leq \frac{\eta kT}{8} + \frac{\log k}{\eta} \ .$$

*Proof.* For any concrete instantiation of $b_0, \ldots, b_{T+1}$, the set $S$ is fixed and the sequence $(\ell_t)_{t \in S}$ is an oblivious sequence of loss functions. Note that the steps performed by Algorithm 1 on the rounds indicated in $S$ are precisely the steps that the standard EXP3 algorithm would perform if it were presented with the loss sequence $(\ell_t)_{t \in S}$. Therefore, Theorem 1 guarantees that

$$\mathbb{E}\left[ \sum_{t \in S} \ell_t(x_t) - \sum_{t \in S} \ell_t(x) \ \middle| \ S \right] \leq \frac{\eta k|S|}{2} + \frac{\log k}{\eta} \ .$$

Taking expectations on both sides of the above and noting that $\mathbb{E}[|S|] \leq T/4$ proves the lemma. $\qquad\square$

Lemma 1 proves a regret bound that is restricted to the rounds indicated by $S$. The following lemma relates that regret to the total regret, on all $T$ rounds.

**Lemma 2.** *For any $x \in \mathcal{K}$, we have*

$$\mathbb{E}\left[ \sum_{t=1}^{T} \ell_t(x_t) \right] - \sum_{t=1}^{T} \ell_t(x) \leq 4 \, \mathbb{E}\left[ \sum_{t \in S} \ell_t(x_t) - \sum_{t \in S} \ell_t(x) \right] + \mathbb{E}\left[ \sum_{t=1}^{T} \|p_t - p_{t-1}\|_1 \right] .$$

*Proof.* Using the definition of $S$, we have

$$\mathbb{E}\left[ \sum_{t \in S} \ell_t(x) \right] = \sum_{t=1}^{T} \ell_t(x) \, \mathbb{E}[(1 - b_t)b_{t+1}] = \frac{1}{4} \sum_{t=1}^{T} \ell_t(x) \ . \tag{2}$$

Similarly, we have

$$\mathbb{E}\left[ \sum_{t \in S} \ell_t(x_t) \right] = \sum_{t=1}^{T} \mathbb{E}\left[ \ell_t(x_t) \, (1 - b_t)b_{t+1} \right] . \tag{3}$$

We focus on the $t$'th summand in the right-hand side above. Since $b_{t+1}$ is independent of $\ell_t(x_t)(1 - b_t)$, it holds that

$$\mathbb{E}\left[ \ell_t(x_t)(1 - b_t)b_{t+1} \right] = \mathbb{E}[b_{t+1}]\mathbb{E}\left[ \ell_t(x_t)(1 - b_t) \right] = \frac{1}{2} \mathbb{E}\left[ \ell_t(x_t)(1 - b_t) \right] \ .$$

Using the law of total expectation, we get

$$\frac{1}{2} \mathbb{E}\left[ \ell_t(x_t)(1 - b_t) \right] = \frac{1}{4} \mathbb{E}\left[ \ell_t(x_t)(1 - b_t) \ \middle| \ b_t = 0 \right] + \frac{1}{4} \mathbb{E}\left[ \ell_t(x_t)(1 - b_t) \ \middle| \ b_t = 1 \right]$$

$$= \frac{1}{4} \mathbb{E}\left[ \ell_t(x_t) \ \middle| \ b_t = 0 \right] \ .$$

If $b_t = 0$ then Algorithm 1 sets $x_t \leftarrow x_{t-1}$ so we have that $x_t = x_{t-1}$. Therefore, the above equals $\frac{1}{4}\mathbb{E}[\ell_t(x_{t-1}) \mid b_t = 0]$. Since $x_{t-1}$ is independent of $b_t$, this simply equals $\frac{1}{4}\mathbb{E}[\ell_t(x_{t-1})]$. Hölder's inequality can be used to upper bound

$$\mathbb{E}[\ell_t(x_t) - \ell_t(x_{t-1})] = \mathbb{E}\Big[\sum_{x \in \mathcal{K}}\big(p_t(x) - p_{t-1}(x)\big)\ell_t(x)\Big] \leq \mathbb{E}[\|p_t - p_{t-1}\|_1] \cdot \max_{x \in \mathcal{K}}\ell_t(x) \;,$$

where we have used the fact that $x_t$ and $x_{t-1}$ are distributed according to $p_t$ and $p_{t-1}$ respectively (regardless of whether an update took place or not). Since it is assumed that $\ell_t(x) \in [0, 1]$ for all $t$ and $x \in \mathcal{K}$, we obtain

$$\frac{1}{4}\mathbb{E}\big[\ell_t(x_{t-1})\big] \geq \frac{1}{4}\big(\mathbb{E}\big[\ell_t(x_t)\big] - \mathbb{E}[\|p_t - p_{t-1}\|_1]\big) \,.$$

Overall, we have shown that

$$\mathbb{E}\big[\ell_t(x_t)(1 - b_t)b_{t+1}\big] \geq \frac{1}{4}\big(\mathbb{E}\big[\ell_t(x_t)\big] - \mathbb{E}[\|p_t - p_{t-1}\|_1]\big) \,.$$

Plugging this inequality back into Eq. (3) gives

$$\mathbb{E}\left[\sum_{t \in S}\ell_t(x_t)\right] \geq \frac{1}{4}\mathbb{E}\left[\sum_{t=1}^{T}\ell_t(x_t) - \sum_{t=1}^{T}\|p_t - p_{t-1}\|_1\right].$$

Summing the inequality above with the one in Eq. (2) concludes the proof. $\qquad\square$

Next, we prove that the probability distributions over arms do not change much on consecutive rounds of EXP3.

**Lemma 3.** *The distributions $p_1, p_2, \ldots, p_T$ generated by the* BLINDED EXP3 *algorithm satisfy* $\mathbb{E}[\|p_{t+1} - p_t\|_1] \leq 2\eta$ *for all $t$.*

*Proof.* Fix a round $t$; we shall prove the stronger claim that $\|p_{t+1} - p_t\|_1 \leq 2\eta$ with probability 1. If no update had occurred on round $t$ and $p_{t+1} = p_t$, this holds trivially. Otherwise, we can use the triangle inequality to bound

$$\|p_{t+1} - p_t\|_1 \leq \|p_{t+1} - w_{t+1}\|_1 + \|w_{t+1} - p_t\|_1 \,,$$

with the vector $w_{t+1}$ as specified in Algorithm 1. Letting $W_{t+1} = \|w_{t+1}\|_1$ we have $p_{t+1} = w_{t+1}/W_{t+1}$, so we can rewrite the first term on the right-hand side above as

$$\|p_{t+1} - W_{t+1} \cdot p_{t+1}\|_1 = |1 - W_{t+1}| \cdot \|p_{t+1}\|_1 = 1 - W_{t+1} = \|p_t - w_{t+1}\|_1 \,,$$

where the last equality follows by observing that $p_t \geq w_{t+1}$ entrywise, $\|p_t\|_1 = 1$ and $\|w_{t+1}\|_1 = W_{t+1}$. By the definition of $w_{t+1}$, the second term on the right-hand side above equals $p_t(x_t) \cdot \big(1 - e^{-\eta\ell_t(x_t)/p_t(x_t)}\big)$. Overall, we have

$$\|p_{t+1} - p_t\|_1 \leq 2p_t(x_t) \cdot \big(1 - e^{-\eta\ell_t(x_t)/p_t(x_t)}\big) \,.$$

Using the inequality $1 - \exp(-\alpha) \leq \alpha$, we get $\|p_{t+1} - p_t\|_1 \leq 2\eta\ell_t(x_t)$. The claim now follows from the assumption that $\ell_t(x_t) \in [0, 1]$. $\qquad\square$

We can now proceed to prove our regret bound.

*Proof of Theorem 2.* Combining the bounds of Lemmas 1–3 proves that for any fixed arm $x \in \mathcal{K}$, it holds that

$$\mathbb{E}\left[\sum_{t=1}^{T}\ell_t(x_t)\right] - \sum_{t=1}^{T}\ell_t(x) \leq \frac{\eta kT}{2} + \frac{4\log k}{\eta} + 2\eta T$$

$$\leq 2\eta kT + \frac{4\log k}{\eta} \,.$$

Specifically, the above holds for the best arm in hindsight. Setting $\eta = \sqrt{\frac{2\log k}{kT}}$ proves the theorem. $\qquad\square$

## 4   Blinded Bandit Linear Optimization

In this section we extend our results to the setting of linear optimization with bandit feedback, formally defined in Section 2. We focus on the GEOMETRICHEDGE algorithm [11], that was the first algorithm for the problem to attain the optimal $O(\sqrt{T})$ regret, and adapt it to the blinded setup.

Our BLINDED GEOMETRICHEDGE algorithm is detailed in Algorithm 2. The algorithm uses a mechanism similar to that of Algorithm 1 for deciding when to avoid switching actions. Following the presentation of [11], we assume that $\mathcal{K} \subseteq [-1, 1]^n$ is finite and that the standard basis vectors $\mathbf{e}_1, \ldots, \mathbf{e}_n$ are contained in $\mathcal{K}$. Then, the set $\mathcal{E} = \{\mathbf{e}_1, \ldots, \mathbf{e}_n\}$ is a barycentric spanner of $\mathcal{K}$ [5] that serves the algorithm as an exploration basis. We denote the uniform distribution over $\mathcal{E}$ by $u_{\mathcal{E}}$.

---

**Algorithm 2:** BLINDED GEOMETRICHEDGE

---

**Parameter:** learning rate $\eta > 0$

let $q_1$ be the uniform distribution over $\mathcal{K}$, and draw $x_0 \sim q_1$
draw $b_0, \ldots, b_{T+1}$ i.i.d. unbiased Bernoullis
set $\gamma \leftarrow n^2 \eta$

**for** $t = 1, 2, \ldots, T$
    set $p_t \leftarrow (1 - \gamma)\, q_t + \gamma\, u_{\mathcal{E}}$
    compute covariance $C_t \leftarrow \mathbb{E}_{x \sim p_t}[xx^\top]$

    **if** $b_{t-1} = 0$ *and* $b_t = 1$
        draw $x_t \sim p_t$                             `// possible switch`
    **else**
        set $x_t \leftarrow x_{t-1}$                           `// no switch`

    play arm $x_t$ and incur loss $\ell_t(x_t) = \ell_t \cdot x_t$

    **if** $b_t = 0$ *and* $b_{t+1} = 1$
        observe $\ell_t(x_t)$ and let $\hat{\ell}_t \leftarrow \ell_t(x_t) \cdot C_t^{-1} x_t$
        update $q_{t+1}(x) \propto q_t(x) \cdot \exp(-\eta \hat{\ell}_t \cdot x)$
    **else**
        set $q_{t+1} \leftarrow q_t$

---

The main result of this section is an $O(\sqrt{T})$ upper-bound over the expected regret of Algorithm 2.

**Theorem 3.** *Let $\ell_1, \ldots, \ell_T$ be an arbitrary sequence of linear loss functions, admissible with respect to the action set $\mathcal{K} \subseteq \mathbb{R}^n$. Let $x_1, \ldots, x_T$ be the random sequence of arms chosen by Algorithm 2 as it plays the blinded bandit game on this sequence, with learning rate fixed to $\eta = \sqrt{\frac{\log(nT)}{10nT}}$. Then,*

$$R(T) \ \leq \ 4n^{3/2}\sqrt{T \log(nT)}\,.$$

With minor modifications, our technique can also be applied to variants of the GEOMETRICHEDGE algorithm (that differ by their exploration basis) for obtaining regret bounds with improved dependence of the dimension $n$. This includes the COMBAND algorithm [8], EXP2 with John's exploration [7], and the more recent version employing volumetric spanners [13].

We now turn to prove Theorem 3. Our first step is proving an analogue of Lemma 1, using the regret bound of the GEOMETRICHEDGE algorithm proved by Dani et al. [11].

**Lemma 4.** *For any $x \in \mathcal{K}$, it holds that $\mathbb{E}\left[\sum_{t \in S} \ell_t(x_t) - \sum_{t \in S} \ell_t(x)\right] \leq \frac{\eta n^2 T}{2} + \frac{n \log(nT)}{2\eta}$.*

We proceed to prove that the distributions generated by Algorithm 2 do not change too quickly.

**Lemma 5.** *The distributions $p_1, p_2, \ldots, p_T$ produced by the BLINDED GEOMETRICHEDGE algorithm (from which the actions $x_1, x_2, \ldots, x_T$ are drawn) satisfy $\mathbb{E}[\|p_{t+1} - p_t\|_1] \leq 4\eta\sqrt{n}$ for all $t$.*

The proofs of both lemmas are omitted due to space constraints. We now prove Theorem 3.

*Proof of Theorem 3.* Notice that the bound of Lemma 2 is independent of the construction of the distributions $p_1, p_2, \ldots, p_T$ and the structure of $\mathcal{K}$, and thus applies for Algorithm 2 as well. Combining this bound with the results of Lemmas 4 and 5, it follows that for any fixed action $x \in \mathcal{K}$,

$$\mathbb{E}\left[\sum_{t=1}^{T} \ell_t(x_t)\right] - \sum_{t=1}^{T} \ell_t(x) \leq \frac{\eta n^2 T}{2} + \frac{n \log(nT)}{2\eta} + 4\eta\sqrt{n}T \leq 5\eta n^2 T + \frac{n \log(nT)}{2\eta} .$$

Setting $\eta = \sqrt{\frac{\log(nT)}{10nT}}$ proves the theorem. $\qquad\square$

## 5 Discussion and Open Problems

In this paper, we studied a new online learning scenario where the player receives feedback from the adversarial environment only when his action is the same as the one from the previous round, a setting that we named *the blinded bandit*. We devised an optimal algorithm for the blinded multi-armed bandit problem based on the EXP3 strategy, and used similar ideas to adapt the GEOMETRICHEDGE algorithm to the blinded bandit linear optimization setting. In fact, a similar analysis can be applied to any online algorithm that does not change its underlying prediction distributions too quickly (in total variation distance).

In the practical examples given in the introduction, where each switch introduces a bias or a variance, we argued that the multi-armed bandit problem with switching costs is an inadequate solution, since it is unreasonable to solve an easy problem by reducing it to one that is substantially harder. Alternatively, one might consider simply ignoring the noise in the feedback after each switch and using a standard adversarial multi-armed bandit algorithm like EXP3 despite the bias or the variance. However, if we do that, the player's observed losses would no longer be oblivious (as the observed loss on round $t$ would depend on $x_{t-1}$), and the regret guarantees of EXP3 would no longer hold[3]. Moreover, any multi-armed bandit algorithm with $O(\sqrt{T})$ regret can be forced to make $\Theta(T)$ switches [12], so the loss observed by the player could actually be non-oblivious in a constant fraction of the rounds, which would deteriorate the performance of EXP3.

Our setting might seem similar to the related problem of label-efficient prediction (with bandit feedback), see [9]. In the label-efficient prediction setting, the feedback for the action performed on some round is received only if the player explicitly asks for it. The player may freely choose when to observe feedback, subject to a global constraint on the number of total feedback queries. In contrast, in our setting there is a strong correlation between the actions the player takes and the presence of the feedback signal. As a consequence, the player is not free to decide when he observes feedback as in the label-efficient setting. Another setting that may seem closely related to our setting is the multi-armed bandit problem with delayed feedback [16, 17]. In this setting, the feedback for the action performed on round $t$ is received at the end of round $t + 1$. However, note that in all of the examples we have discussed, the feedback is always immediate, but is either nonexistent or unreliable right after a switch. The important aspect of our setup, which does not apply to the label-efficient and delayed feedback settings, is that the feedback adapts to the player's past actions.

Our work leaves a few interesting questions for future research. A closely related adaptive-feedback problem is one where feedback is revealed only on rounds where the player *does* switch actions. Can the player attain $O(\sqrt{T})$ regret in this setting as well, or is the need to constantly switch actions detrimental to the player? More generally, we can consider other multi-armed bandit problems with adaptive feedback, where the feedback depends on the player's actions on previous rounds. It would be quite interesting to understand what kind of adaptive-feedback patterns give rise to easy problems, for which a regret of $O(\sqrt{T})$ is attainable. Specifically, is there a problem with oblivious losses and adaptive feedback whose minimax regret is $\Theta(T^{2/3})$, as is the case with adaptive losses?

**Acknowledgments**

The research leading to these results has received funding from the Microsoft-Technion EC center, and the European Union's Seventh Framework Programme (FP7/2007-2013]) under grant agreement n° 336078 ERC-SUBLRN.

## Footnotes

[1]The classification of online problems into *easy* vs. *hard* is borrowed from Antos et al. [2].

[2]More generally, we could define a setting where the player is blinded for $m$ rounds following each switch, but for simplicity we focus on $m = 1$.

[3]Auer et al. [4] also present an algorithm called EXP3.P and seemingly prove $O(\sqrt{T})$ regret guarantees against non-oblivious adversaries. These bounds are irrelevant in our setting—see Arora et al. [3].

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
