[Reviews · NeurIPS 2014]

Submitted by Assigned_Reviewer_5

This paper introduces the following version of the multi-armed bandit problem: The learner gets feedback (the loss of the chosen arm) only when the chosen arm is the same as the one chosen in the previous round. That is, whenever the learner changes arms, he does not receive any feedback. The paper offers algorithms and analyses for this modified version of MAB as well as for the extension to bandit linear optimization. The bounds are asymptotically identical to those of MAB.

The paper is very well written, the work is polished, the algorithm is a clever modification of the standard Exp3 algorithm. I checked the proofs in Section 3, they seem to be correct.

One thing that puzzles me is the question of why we need the extra Bernoulli sequence for the decision of changing arms. I would imagine the algorithm would also work by just saying that we choose every action twice in a row before choosing a new one. If there is a problem with this rule not being randomized (and thus the adversary can choose tricky losses to get linear regret), let's say we draw one unbiased Bernoulli in the beginning to decide if we switch arms on even or on odd time steps.
Summary: Solid work on a modified MAB model, well written, clever algorithm that builds on standard Exp3.

Submitted by Assigned_Reviewer_9

The paper studies a nice variant of non-stochastic K-armed bandit problem and linear bandit optimization, in which feedback (about the reward) is not obtained in rounds where the algorithm changes action (compared to the previous round). The paper presents a clever modification of standard EXP3 (from Auer et al) algorithm and proves upper bound on the regret of the modified algorithm. The regret bound happens to coincide with the standard bounds, which is somewhat surprising, since in previously studied model, where the learner pays for the action switches, the regret bounds changes completely.

The paper is clearly written, the ideas are new and original. The proofs seem correct. I am not sure how significant this particular variant of the multi-armed problem is from practical perspective. (The paper gives states several potential applications in the introduction.) In any case, from theoretical perspective this is an important problem, since it demonstrates that even a non-trivially crippled learner can still achieve optimal learning rate.

I am happy to recommend the paper for acceptance at NIPS.
Summary: Paper studies a nice modification of the multi-armed bandit problem, where the learner doesn't receive feedback in the round in which switch of action occurred. The paper gives an algorithm and proves upper bounds on its regret, which surprisingly matches the upper bound for the classical problem. I recommend the paper for acceptance at NIPS.

Submitted by Assigned_Reviewer_19

The paper considers an online learning game with a special sort of partial information, where the learner only gets bandit feedback on its loss whenever it sticks with its previously chosen action. The authors show the surprising fact that in this so-called blinded bandit setting, it is still possible to attain regret guarantees of the same order as in the standard multi-armed bandit setting by using a simple modification of the classic Exp3 strategy. The results are also extended to the case of bandit linear optimization.

Originality & Significance
--------------------------
In light of recent results by Dekel et al., it is indeed surprising that this learning setting is no more difficult than the standard bandit setting. Even though I wouldn't call this observation groundbreaking, it still constitutes an interesting contribution to our general understanding of the role of information in online learning. The algorithm itself and its analysis are quite elegant: the approach is based on a simple and clever subsampling strategy of the stream of observations that effectively separates in time the selection of actions and observing the associated losses. While specifically tailored to this application, this strategy can be possibly applied in other online learning problems with delayed feedback.

Quality & Clarity
-----------------
The paper is written very well and the technical quality is very good, apart from a few minor technical issues. Precisely, I have located one bit in the analysis that needs to be fixed.

Expectations and conditional expectations are treated quite liberally throughout the paper. The first occurrence of this problem is to be found near line 264, where the authors use the fact that "b_{t+1}" is independent of l_t(x_t)(1-b_t)", which of course only holds when conditioned on the history up to time t. While this specific problem doesn't stir up any trouble, the same bug shows up on line 274, where the distributions of x_t and x_{t-1} are p_{t} and p_{t-1}, respectively, but under different conditions. Treating this issue complicates the analysis a bit (by an additional application of the tower rule and Jensen's inequality), but it seems that the analysis still goes through -- although the authors should provide a precise proof before the paper can be accepted.

Detailed comments
-----------------
065: The example in this paragraph, while sensible, is probably not the best fit for this framework, as the increased activity *does* influence the true rewards, not only their observations -- the two are closely tied together. Either remove this example or elaborate if you still think it's correct.
169: "maintains a probability" -> "maintains a probability distribution"
172: p_1 -> p_t
297: "The claim is trivial for rounds in which no update occurred" -- This statement is difficult to interpret, as the claim is about an expectation that integrates over the randomness inducing the updates. That is, it is meaningless to refer to action switches unless one conditions on the sequence of b_t's. Please fix this.
388: Note that the last inequality follows from n \ge 2.
332: The result of Dani et al. is proven for continuous decision spaces by laying an appropriately-sized grid on the decision space and running GeometricHedge on the resulting discrete action set. That is, their algorithm does not match the one presented as Algorithm 2. If you can prove Lemma 4 for this variant of the algorithm, please include it, otherwise, state that the decision space should be finite. In the latter case, the integrals in the proof of Lemma 5 should be replaced by sums.
Summary: The paper proposes a simple and elegant algorithm for an interesting problem setting that hasn't been considered in previous literature. The results are surprising enough to warrant acceptance.
Author Feedback
Author rebuttal: We thank the reviewers for their thoughtful comments and suggestions.

Reviewer 19:
Thanks for checking the proofs so carefully and for pointing out the minor mistakes. We have confirmed that the bugs are all easily fixable and the analysis in the final version will be rock solid. We will follow all of your suggestions. Thanks again.

Reviewer 5:
The idea of flipping a single coin to choose between odd and even switches will not work. First, if we deterministically choose to switch only on odd rounds, you are correct that the adversary can exploit this to inflict a linear regret. If we flip a coin to choose between odd and even, we still play "odd" with probability 1/2. The adversary simply assumes that we will choose "odd": the expected regret will be linear with probability 1/2 and non-negative with probability 1/2, which is linear overall.

Reviewer 9:
Thanks!